



**Declines and peaks in NO₂ pollution during the multiple waves**
**of the COVID-19 pandemic in the New York metropolitan area**
Maria Tzortziou [1,2], Charlotte F. Kwong [1], Daniel Goldberg [3], Luke Schiferl [4], Róisín
Commane[4,5], Nader Abuhassan [2,6], James J. Szykman [7,8], Lukas C. Valin [8]
[1] Center for Discovery and Innovation, Earth & Atmospheric Sciences, City College of New York, New York, NY
10031, USA
[2] NASA Goddard Space Flight Center, Greenbelt, MD, 20771, USA
[3] Department of Environmental and Occupational Health, George Washington University, Washington, DC, 20052,
USA
[4] Lamont Doherty Earth Observatory, Columbia University, Palisades, NY, 10964, USA
[5] Department of Earth and Environmental Sciences, Columbia University, New York, NY, 10027, USA
[6] Joint Center for Earth Systems Technology, University of Maryland, Baltimore, MD, 21201, USA
[7] NASA Langley Research Center, Hampton, VA, 23666, USA
[8] US EPA/Office of Research and Development/Center for Environmental Measurement and Modeling, Research
Triangle Park, NC, USA
*Correspondence to*: Maria Tzortziou (mtzortziou@ccny.cuny.edu)
ORCID
MT: 0000-0002-4510-7827
CK: 0000-0002-5977-4532
DG: 0000-0003-0784-3986
LDS: 0000-0002-5047-2490
RC: 0000-0003-1373-1550
LV: 0000-0002-7314-3868



**Abstract.** The COVID-19 pandemic created an extreme natural experiment in which sudden changes in human behavior and economic activity resulted in significant declines in nitrogen oxide (NOx) emissions, immediately after strict lockdowns were imposed. Here we examined the impact of multiple waves and response phases of the pandemic on nitrogen dioxide ($NO_2$) dynamics and the role of meteorology in shaping relative contributions from different emission sectors to $NO_2$ pollution in post-pandemic New York City. Long term (> 3.5 years), high frequency measurements from a network of ground-based Pandora spectrometers were combined with TROPOMI satellite retrievals, meteorological data, mobility trends, and atmospheric transport model simulations to quantify changes in $NO_2$ across the New York metropolitan area. The stringent lockdown measures after the first pandemic wave resulted in a decline in top-down NOx emissions by approx. 30% on top of long-term trends, in agreement with sector-specific changes in NOx emissions. Ground-based measurements showed a sudden drop in total column $NO_2$ in spring 2020, by up to 36% in Manhattan and 19-29% in Queens, New Jersey and Connecticut, and a clear weakening (by 16%) of the typical weekly $NO_2$ cycle. Extending our analysis to more than a year after the initial lockdown captured a gradual recovery in $NO_2$ across the NY/NJ/CT tri-state area in summer and fall 2020, as social restrictions eased, followed by a second decline in $NO_2$ coincident with the second wave of the pandemic and resurgence of lockdown measures in winter 2021. Meteorology was not found to have a strong $NO_2$ biasing effect in New York City after the first pandemic wave. Winds, however, were favorable for low $NO_2$ conditions in Manhattan during the second wave of the pandemic, resulting in larger column $NO_2$ declines than expected based on changes in transportation emissions alone. Meteorology played a key role in shaping the relative contributions from different emission sectors to $NO_2$ pollution in the city, with low-speed (< 5 $ms^{-1}$) SW-SE winds enhancing contributions from the high-emitting power-generation sector in NJ and Queens and driving particularly high $NO_2$ pollution episodes in Manhattan, even during – and despite - the stringent early lockdowns. These results have important implications for air quality management in New York City, and highlight the value of high resolution $NO_2$ measurements in assessing the effects of rapid meteorological changes on air quality conditions and the effectiveness of sector-specific NOx emission control strategies.



## 1. Introduction

The global outbreak of the Coronavirus Disease 2019 (COVID-19) profoundly changed the world. From school closures to remote work and other physical distancing measures, this crisis changed the way we move within our communities, potentially with long term implications (Barbieri et al., 2021; Przybylowski et al., 2021). Altered mobility patterns led to sudden and significant worldwide decreases in nitrogen oxide (NOx) emissions from the transportation sector, as documented in many studies focusing on air quality changes immediately after the initial lockdowns (e.g., Liu et al., 2020; Goldberg et al., 2020; Gkatzelis et al., 2021). Yet, the impact of multiple pandemic waves over longer time periods, and the role of meteorology and sector-specific emissions as key drivers of high NOx pollution episodes that occurred in major cities such as New York - even during, and despite, the most stringent early lockdown periods - remain largely unknown, driving this study.

New York City, the most populous and most densely populated city in the Unites States, was hit particularly hard by the pandemic. By late-March 2020, the tri-state region of New York (NY), New Jersey (NJ) and Connecticut (CT) declared a disaster emergency and issued stay-at-home restrictions in response to COVID-19. Almost 8 million New Yorkers sheltered-in-place, while roughly 5% of New York City residents (about 420,000 people) left the city between March and May (Quealy, 2020; Bounds, 2020). The largest decrease in residential population occurred in Manhattan−with more than 30% reduction in relatively wealthy neighborhoods including Upper West and Upper East Side−while the rest of the city saw comparably modest losses (Quealy, 2020). The entire New York metropolitan area remained in lockdown with strict social distancing measures, including school and non-essential business closures, limited transit services, and suspension of public events and gatherings, for more than two months, from mid-March through June 2020. Lockdown measures were relaxed and the first phase of reopening began in June with the area progressing to the final stage of reopening in July. Yet, social distancing measures became strict again, including school closures, as the city experienced a surge in COVID-19 cases in late fall 2020 that reached a maximum in mid-January 2021 with more reported cases to NYC Department of Health and Mental Hygiene than during the first wave of the pandemic (Fig. S1). Early studies using satellite data from the Ozone Monitoring Instrument (OMI) and the Tropospheric Monitoring Instrument (TROPOMI) revealed 31(±14)% and 28(±11)% reduction, respectively, in nitrogen dioxide ($NO_2$) column amount within a 100-km radius of New York City during the three weeks following the onset of the pandemic compared to the same period in 2019 (Bauwens et al., 2020). Similarly, Goldberg et al., (2020) reported a 20% drop in TROPOMI $NO_2$ within a 0.4° radius of New York between March 13 and April 30, 2020.

Emitted to the atmosphere primarily during fossil fuel combustion, nitrogen oxides ($NOx=NO+NO_2$) are a major source of air pollution and necessary precursors of tropospheric ozone, impacting climate as well as human and ecosystem health (Fares et al., 2013; Duan et al., 2019; Lim et al., 2012; Burnett et al., 2004). High $NO_2$ levels have been associated with lung irritation and reduced lung function, increased asthma attacks, cardiovascular disorders, as well as lower birth weight in newborns and increased risk of premature death (U.S. EPA 2016). In addition, through wet and dry deposition, the atmosphere is a major source of excess nitrogen to many terrestrial and aquatic ecosystems



worldwide (Paerl et al., 2002; Pardo et al., 2011). Prior studies have indicated atmospheric deposition accounts for a third or more of total nitrogen loading in systems such as the Chesapeake Bay and Long Island Sound, with important implications for soil biogeochemistry, aquatic biology, development of coastal eutrophication, harmful algal blooms, and hypoxia (e.g., Stacey et al., 2001; Decina et al., 2017; Decina et al., 2020). A combination of strict air quality regulation policies (e.g., Clean Air Interstate Rule, CAIR, 2009) and technological improvements over the past two decades has resulted in significant declines in NOx emissions over the continental United States (van der A et al., 2008; Duncan et al., 2016; Krotkov et al., 2016). Satellite Aura/OMI observations have captured an approximately 4% yr$^{-1}$ decrease in column $NO_2$ levels between 2005 and 2015 over the eastern United States (Krotkov et al., 2016) and a 46% decline in $NO_X$ emissions has been reported for New York City over the period from 2006 to 2017 (Goldberg et al., 2019a). Despite these improvements, air pollution continues to be the single biggest environmental health risk in the United States and globally today (Burnett et al., 2018; Thakrar et al., 2020; WHO 2019). With significant NOx emissions from various sectors (e.g., transportation, energy, industrial), the New York metropolitan area experiences among the highest national $NO_2$ levels (Herman et al., 2018) and has the worst nonattainment record of ozone in eastern North America (based on the EPA 2015 standard) (Karambelas et al., 2020).

Restrictions on human and economic activities, particularly reductions in transportation emissions due to the COVID-19 stay-at-home orders, provide a unique opportunity to assess the importance of different sources of air pollution in New York City and how further sector-specific NOx emission reductions may impact nitrogen pollution in this major urban center. The overarching objective of this study was to examine how $NO_2$ dynamics changed in the New York metropolitan area during the multiple phases of the pandemic and across regions experiencing different shifts in mobility patterns. Ground-based measurements conducted over a period of 3.5 years (2017-2021) allowed us to capture inter-annual variability, impacts of meteorology, and changes in air quality as human behavior changed during the multiple pandemic waves and as vehicle traffic started to return to near pre-pandemic levels a year after the initial lockdown. Combining these high-frequency observations with model simulations and satellite imagery uniquely captured $NO_2$ dynamics across multiple scales and highlighted the impact of COVID-19 restrictions not only on $NO_2$ column amounts but also on $NO_2$ spatiotemporal behavior, including seasonal and weekly cycles.

Meteorological factors have a significant impact on atmospheric chemistry as well as transport, transformation, and dispersion of air pollutants (Xu et al., 2011; Banta et al., 2011; Goldberg et al., 2020). Elucidating the role of meteorology is thus important in assessments of COVID-19 impacts on urban air quality (Gkatzelis et al., 2021). Seasonality and local meteorology were previously reported to drive $NO_2$ changes in New York City as large as a factor of two over the course of a year (Goldberg et al., 2020). Although meteorological patterns were especially favorable for low $NO_2$ in much of the United States in spring 2020, varying meteorological conditions in New York City were not found to have a biasing effect in TROPOMI estimates of $NO_2$ declines during the initial lockdown period (Goldberg et al., 2020). Because our study extended over a longer time-period, we explicitly investigated how weather conditions may have impacted observed changes in $NO_2$ pollution and the relative contribution of different $NO_x$ emissions sectors (i.e., energy versus transportation) during the multiple phases of the pandemic.





## 2. Methods

### 2.1 Ground-based measurements of column NO₂ dynamics

To assess the impact of COVID-19 restrictions on $NO_2$ spatiotemporal behavior we used high-frequency measurements of total column $NO_2$ ($TCNO_2$) from the ground-based Pandonia Global Network (PGN, https://www.pandonia-global-network.org/). Sponsored by the National Aeronautics and Space Administration (NASA) and the European Space Agency (ESA), PGN focuses on providing long-term, real-time and verified QA/QC data on air quality and atmospheric composition from a network of standardized and calibrated Pandora spectrometer instruments (PSIs, Herman et al., 2019). The PGN global network serves as a validation resource for UV-visible satellite sensors on low-earth and geostationary orbit, and recent studies have included Pandora measurements for ground-based validation of TROPOMI $NO_2$ measurements near New York City and Long Island Sound (Judd et al., 2020; Verhoelst et al., 2021). In the New York metropolitan area, PGN sites include Manhattan, NY (PSI #135), Queens, NY (PSIs #55, #140), New Brunswick, NJ (PSIs #56, #69), and New Haven, CT (PSIs #20, #64) (Table 1, Fig. 1). PSI #135 in Upper West Manhattan, NY, has the longest data record (since Dec 2017) among these instruments and is located on the Advanced Science Research Center (ASRC) Rooftop Observatory at the City College of New York campus, an intensive urban air-quality monitoring site. The Pandora sensor in Queens, NY, is located at the CUNY Queens College, a New York Department of Environmental Conservation (NYDEC) Air Toxics and NCore

| Pandora name, #, location (Principal Investigator) | Temporal range of data | | TCNO₂ (in DU) | | | | | | | |
|---|---|---|---|---|---|---|---|---|---|---|
| | | | Apr-May | | June-Aug | | Sept-Nov | | Dec-Feb | |
| | | | Pre- | Post- | Pre- | Post- | Pre- | Post- | Pre- | Post- |
| Manhattan, NY PSI #135 40.8153°, -73.9505° | 12/2017 - Present | mean | 0.61 | 0.39 | 0.59 | 0.44 | 0.59 | 0.46 | 0.71 | 0.48 |
| | | stdev | 0.34 | 0.25 | 0.35 | 0.24 | 0.38 | 0.27 | 0.45 | 0.30 |
| | | max | 3.11 | 3.25 | 3.77 | 2.09 | 2.94 | 1.89 | 3.13 | 2.05 |
| (M. Tzortziou) | | **change** | | -36% | | -25% | | -22% | | -32% |
| Queens, NY PSI #140, #55 40.7361°, -73.8215° | 5/2018 - Present | mean | 0.61 | 0.48 | 0.54 | 0.51 | 0.57 | 0.51 | 0.73 | 0.70 |
| | | stdev | 0.35 | 0.21 | 0.28 | 0.19 | 0.33 | 0.22 | 0.40 | 0.38 |
| | | max | 3.42 | 3.60 | 2.74 | 1.54 | 3.36 | 2.34 | 3.04 | 2.81 |
| (J. Szykman) | | **change** | | -21% | | -6% | | -11% | | -4% |
| New Brunswick, NJ* PSI #56, #69 40.4622°, -74.4294° | 5/2018 - 1/2021 | mean | 0.32 | 0.26 | 0.29 | 0.28 | 0.34 | 0.30 | 0.42 | 0.26 |
| | | stdev | 0.15 | 0.18 | 0.15 | 0.20 | 0.24x | 0.21 | 0.31 | 0.10 |
| | | max | 1.46 | 2.06 | 1.98 | 2.42 | 2.55 | 4.59 | 2.72 | 0.53 |
| (J. Szykman) | | **change** | | -19% | | -3% | | -12% | | -38% |
| New Haven, CT PSI #20, #64 41.3014°, -72.9029° | 5/2018 - Present | mean | 0.38 | 0.27 | 0.34 | 0.29 | 0.34 | 0.29 | 0.36 | 0.33 |
| | | stdev | 0.11 | 0.08 | 0.09 | 0.08 | 0.15 | 0.13 | 0.17 | 0.18 |
| | | max | 0.75 | 0.78 | 0.77 | 0.83 | 1.71 | 1.13 | 1.37 | 1.83 |
| (J. Szykman) | | **change** | | -29% | | -15% | | -15% | | -8% |

\* The Dec – Feb period for New Brunswick contains 16 days of data in December 2020, 1 day of data in January 2021, and no data in February 2021.

**Table 1: Pandora sites (including names of Local Principal Investigator (PI)), and mean, standard deviation (stdev) and maximum (max) total column NO₂ (TCNO₂) amounts (based on half-hour averages) measured pre- and post- the COVID-19 lockdown in New York.**



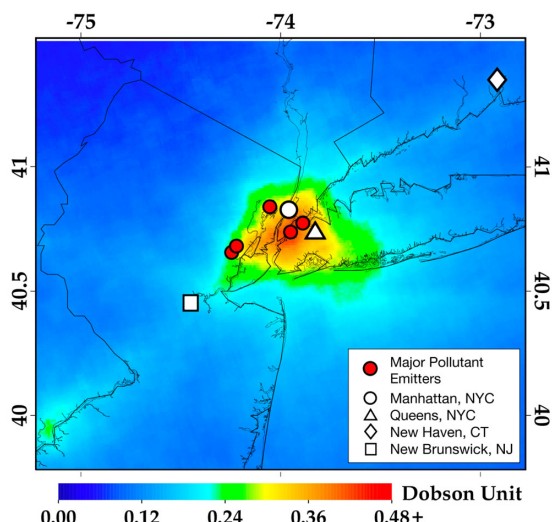

**Figure 1: Map of study area, indicating location of Pandora sensors (white symbols) in Manhattan NY, Queens NY, New Brunswick NJ, and New Haven CT, overlaid with mean 2019 annual total column NO₂ from TROPOMI (in DU). Major pollutant emitters (red circles) in the area are included, specifically the PSEG Bergen Generating Station in Ridgefield, and the Linden Generating Station, the Linden Cogeneration Facility, and the Phillips 66 Bayway Refinery (major emission sources in NJ), and the Astoria and Ravenswood Generating Stations in Queens, NY (among the largest greenhouse gas polluters in the state of NY in 2018 and 2019).**

monitoring site within a dense residential neighborhood and near several major roadways. The Pandora in New Haven,
CT, is located at the Connecticut Department of Energy and Environmental Protection (CTDEEP) Photochemical
Assessment Monitoring Station (PAMS) in Criscuolo Park, at the confluence of the Mill and Quinnipiac Rivers
surrounded by a residential neighborhood near the elevated intersection of three major highways and industrial
activities across the rivers. The New Jersey Department of Environmental Protection (NJDEP) Photochemical
Assessment Monitoring Station (PAMS) in New Brunswick, NJ, includes a Pandora sensor located on the roof of the
Rutgers (NJDEP) research shelter dedicated to atmospheric research, on a university research farm in a suburban
neighborhood and approximately 20 km from the coast.

Pandora is a sun/sky/lunar passive UV/Visible spectrometer system, driven by a highly accurate sun tracker that points
an optical head at the sun and transmits the received light to an Avantes low stray light CCD spectrometer (spectral
range: 280-525 nm; spectral resolution: 0.6 nm with 4 times oversampling) through a fiber optic cable (Herman et al.,
2019; Tzortziou et al., 2014). The spectrometer is temperature stabilized at 20°C inside a weather resistant container.
Trace gas abundances along the light path are determined using differential optical absorption spectroscopy (DOAS).
The system can operate in both direct-sun and sky-scan mode for retrievals of $O_3$, $NO_2$, $SO_2$ and $CH_2O$ total columns,
tropospheric columns, and information on vertical profile (Tzortziou et al., 2018; Herman et al., 2018; Spinei et al.,
2018), and is an enhanced monitoring instrument for characterizing upper air pollutants under the U.S. EPA PAMS
program (Szykman et al., 2019). The estimated $TCNO_2$ error in Pandora retrievals is approximately 0.05 DU (1 DU



= $2.69 \times 10^{16}$ molecules $cm^{-2}$) (Herman et al., 2019). Pandora data were filtered here for normalized root-mean square
of weighted spectral fitting residuals less than 0.05, uncertainty in $NO_2$ retrievals less than 0.05 DU, and $TCNO_2 > 0$.

### 2.2 TROPOMI satellite retrievals

Jointly developed by the Netherlands and ESA, TROPOMI is an air quality monitoring sensor onboard the sun-
synchronous Copernicus Sentinel-5 Precursor satellite, launched on 13 October 2017 (Veefkind et al., 2012). On a
low-earth (825 km) orbit, Sentinel-5P has a daily equator overpass time of approximately 13:30 local time and global
daily coverage. TROPOMI has a spatial resolution of 7.2 km (5.6 km as of 6 August 2019) along-track by 3.6 km
across-track at nadir, a significant improvement compared to its predecessors OMI (Ozone Monitoring Instrument)
and SCIAMACHY (SCanning Imaging Absorption spectroMeter for Atmospheric CartograpHY). Several spectral
bands in the ultraviolet to shortwave-infrared (270-2385 nm) and a spectral resolution between 0.25 and 1 nm, allow
observations of cloud, aerosol properties, and key atmospheric trace gases including $O_3$, $NO_2$, $CO$, $SO_2$, $CH_4$ and
$CH_2O$ (Veefkind et al., 2012). $NO_2$ retrievals from TROPOMI are based on measurements in the 405–465 nm spectral
window. Using a DOAS technique, similar to the Pandora instrument, the top-of-atmosphere spectral radiances are
converted into slant column amounts of $NO_2$ between the sensor and the Earth's surface (Boersma et al., 2018). In two
additional steps, subtraction of the stratospheric component and incorporation of an air mass factor, the slant column
quantity is converted into a tropospheric vertical column content (Beirle et al., 2019; Dix et al., 2020; Goldberg et al.,
2019b; Griffin et al., 2019; Ialongo et al., 2020; Reuter et al., 2019; Zhao et al., 2020). For this analysis, we used the
operational "off-line" TROPOMI $NO_2$ data set, Version 1.02 between 30 April 2018 – 19 March 2019 and Version
1.03 20 March 2019 – 28 November 2020.  We do not continue the TROPOMI analysis beyond 28 November 2020
due to a significant change in the algorithm (to version 1.04) on 29 November 2020. TROPOMI data are filtered using
a quality assurance flag (QA), in which pixels with QA values greater than 0.75 are utilized; no other filter has been
applied. Validation of TROPOMI $NO_2$ V1.2 tropospheric columns over the New York City metropolitan area indicate
columns are biased low, varying 19-33% (Judd et al., 2020).

### 2.3 Satellite-derived NOx emissions

We used an inverse statistical modeling technique (Goldberg et al., 2019b; Laughner & Cohen 2019) to derive the
New York City $NO_X$ emission rates from a combination of TROPOMI satellite data and re-analysis meteorology. This
method accounts for daily changes in temperature, sun angle, wind speed and wind direction by calculating a
spatiotemporally specific $NO_2$ lifetime. In brief, all $NO_2$ satellite data over New York City were compiled and rotated
based on the daily-observed wind direction, so that the oversampled plume is decaying in a single direction. We used
the closest gridded value without interpolation of the 100-m (above the surface) horizontal wind speed and direction
from the ERA5 re-analysis dataset (Hersbach et al., 2020) generated at 0.25º × 0.25º. Once all daily plumes were
rotated to be aligned as an effective horizontal plume and averaged together during a 5-month warm season period
(May-Sept; usually ~75 snapshots), we integrated ±0.5º along the y-axis about the x-axis to compute a one-



dimensional line density in units of mass per distance. The line densities, which are parallel to the wind direction,
peak near the primary $NO_X$ emissions source and gradually decay downwind from a combination of atmospheric
dispersion, chemical transformation, and deposition. The line densities were fit to a statistical exponentially modified
Gaussian (EMG) model (Beirle et al., 2011; de Foy et al., 2014; Valin et al., 2013; Verstraeten et al., 2018). The five
fitted parameters of the statistical fit are the $NO_2$ background, $NO_2$ mass perturbed above the background threshold
(burden), decay distance, horizontal location of apparent source (ideally at the origin), and sigma of the Gaussian
plume. The $NO_X$ emissions rate from the source can be calculated from the $NO_2$ burden, decay distance, and $NO_X/NO_2$
ratio, which previous work has shown to be 1.33 (Beirle et al., 2011). After accounting for a systematic low bias of
TROPOMI in polluted areas (Judd et al., 2020; Verhoelst et al., 2021), the $NO_X$ emissions compare well with known
emissions from power plants (Goldberg et al., 2019b). For this project, we do not correct for TROPOMI low bias, but
instead assume the low bias is consistent between years and calculate changes between years. A full description of the
method can be found in Goldberg et al. (2019a; b).
**2.4 STILT model simulations**
We used STILT, the Stochastic Time-Inverted Lagrangian Transport model, to calculate the surface influence and
contributions from different sources of $NO_2$ pollution to the city. STILT is a Lagrangian particle dispersion model, in
this case driven by NOAA High-Resolution Rapid Refresh (HRRR) meteorology at 3 km horizontal resolution, that
follows the trajectory of 500 air parcels released from the receptor (measurement site) position backward in time over
the previous 24 hours. The motion of each parcel is determined by both advection by the large-scale wind fields and
random turbulent motion, independent of the other parcels. The proportion of parcels residing in the lower half of the
planetary boundary layer determines the influence of surface fluxes on the measured mole fractions. This surface
influence is tracked in time and space, which allows for the calculation of a two-dimensional footprint at hourly
intervals over the travel period and spatial domain of the particles. The unit of surface influence is defined as the
response of each receptor concentration measurement to a unit emission of a trace gas at each grid square (e.g., ppb
($\mu mol\ m^{-2}s^{-1})^{-1}$). In this study, we ran hourly STILT simulations for the 10 hours surrounding daily peak $NO_2$, for cases
of particularly high column $NO_2$ amounts (> 1.8 DU, more than three times the average of pre-pandemic levels)
measured at the Manhattan and Queens Pandora sites during the COVID-19 lockdown in April 2020 and after the
shutdown in October 2020. Simulated particles originated at the elevation of the Pandora instruments. We also
performed simulations for one low $NO_2$ case in April 2020 for comparison. The STILT footprints were multiplied by
2015 annual gridded maps of $NO_x$ emissions ($\mu mol\ m^{-2}s^{-1}$) at 0.1° horizonal resolution from the Emissions Database
for Global Atmospheric Research (EDGAR) v5.0, which combine atmospheric pollutant data categorized by
anthropogenic emissions sector (e.g., power, manufacturing, transportation), to predict the $NO_2$ concentration
enhancement (ppb) that would be expected for each observed hour.





**2.5 Meteorological Data**

Wind speed and direction data from the ERA5 Model (Copernicus Climate Change Service (C3S), 2017) were used to examine the impact of meteorology on TROPOMI retrieved $NO_2$ column amounts. To downscale the $0.25° \times 0.25°$ grid ERA5 reanalysis, we spatially interpolate daily averaged winds to $0.01° \times 0.01°$ using bilinear interpolation (Goldberg et al., 2020). The average 100-m winds during 16–21 UTC (i.e., approximately the TROPOMI overpass time over North America) were used in our analysis. To assess impacts of meteorology on ground-based measurements of $TCNO_2$ from the Manhattan Pandora PSI#135, we used in situ measurements of wind speed and wind direction (measured at a resolution of 0.01 m/s and 1°, respectively) collected by a collocated ATMOS 41 All-In-One weather station on a 15-minute timescale.

**2.6 Calculation of change in $NO_2$ column amounts**

Change in $NO_2$ column amounts was estimated by comparing post-lockdown TROPOMI and Pandora measurements to the same timeframe in 2018-2019, to account for seasonality and interannual variability (Goldberg et al., 2020; Bauwens et al., 2020). The impact of meteorology on these estimates was explicitly quantified using ERA5 and in situ meteorological data. We estimated changes in $NO_2$ over the different phases of the pandemic in New York City (i) immediately following the initial lockdown in April-May 2020, (ii) as restrictions gradually eased in June-August 2020, (iii) during the re-opening phase in September-November 2020, (iv) as restriction became strict again in December 2020-February 2021 due to the second wave of the pandemic, and (v) in March-April 2021, one year after the initial lockdown. Pandora data were first averaged in half-hour bins to eliminate bias towards times of day with more data, then averaged on weekly, monthly, and seasonal time scales. To examine weekly cycles from satellite observations, TROPOMI data were averaged over longer timescales (April-November), due to the lower temporal resolution and impacts of clouds on satellite retrievals. All computed means for seasonal and weekly cycles were calculated with 95% confidence intervals using a two-tailed single sample t-test. While $NO_2$ data is non-normally distributed, all sample sizes are large (n > 100), and statistics (e.g., p-values) were also calculated using the nonparametric Mann-Whitney and Kruskal Wallis tests which confirmed the validity of t-test results.

**2.7 Changes in mobility patterns**

To examine changes in mobility patterns, we looked at sector-specific mobility indices provided by Apple (Forster et al., 2020) and traffic counts from the Metropolitan Transport Authority (MTA) day-by-day transit data, focusing on bridge and tunnel ridership to represent passenger vehicles (buses, motorcycles, cars, trucks) (NY MTA). Apple mobility data (accessed on 4 June 2021) tracked mobile phone movements and compared post-COVID-19 data with the average on February 13, 2020 (Forster et al., 2020). For MTA data (accessed on 4 June 2021), bridge and tunnel traffic was quantified from E-ZPass and cash toll collection, and percent (%) changes in ridership were calculated through comparison to traffic on the pre-COVID equivalent day in the previous year.





## 3. Results and Discussion

### 3.1. Changes in NO₂ column amounts and spatiotemporal dynamics

Satellite imagery from TROPOMI captured significant post-shutdown $NO_2$ reductions in the New York metropolitan area, particularly during the first three months after the initial lockdowns (Fig. 2). As MTA bridge and tunnel traffic plummeted by up to 80% in April 2020 (Fig. S2), $TCNO_2$ over a 50 x 50 km area around Manhattan dropped by 32% in March-May 2020 compared to the same period in 2018-2019 (Fig. 2, left panel). Smaller declines (< 30%) were found in the surrounding areas of NJ, upstate NY, and CT. These results are consistent with Bauwens et al., (2020) reporting a decline in TROPOMI $NO_2$ column by 28(±11)% within a 100-km radius around New York City during the three weeks following the onset of the pandemic compared to the same period in 2019. By June-August 2020, total $NO_2$ columns−lower during summer due to increased photochemical loss−rose closer to pre-pandemic levels, with approx. 15% decline over New York City and even smaller changes (<10%) in western NJ, CT, and eastern Long Island (Fig. 2, mid panel). This recovery in $NO_2$ coincided with the city of New York commencing the first phase of its reopening plan in June 2020 and gradually relaxing lockdown measures, including the opening of restaurants (outdoor dining) and some workplaces. Daily traffic on New York City bridges and tunnels increased to 22% lower than baseline in summer 2020 (Fig. S2). This trend continued in fall 2020, with $TCNO_2$ showing 13% drop over New York City and smaller declines over more rural areas in northern NJ and eastern Long Island (Fig. 2, right panel).

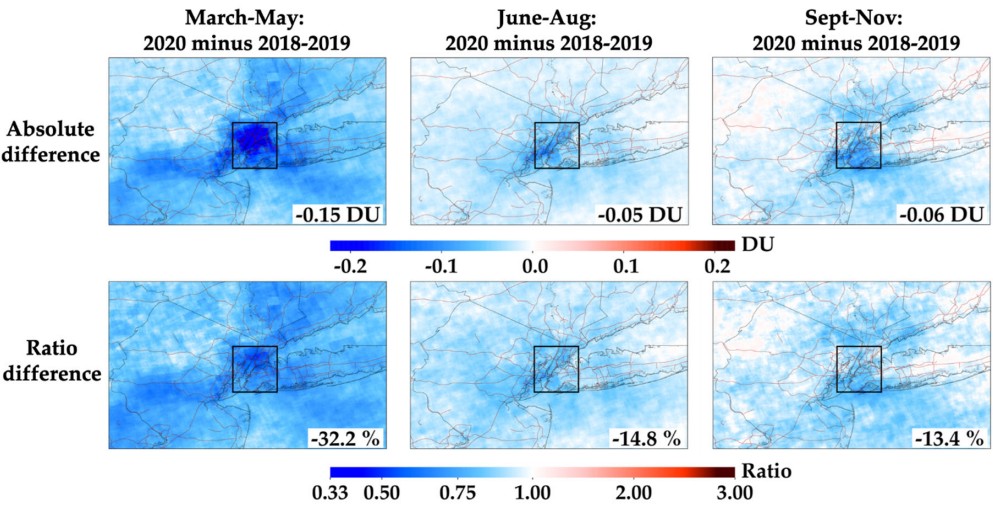

**Figure 2: TROPOMI total vertical column NO₂ differences between 2018-2019 and 2020, over the New York metropolitan area. Results are shown for 13 March through May (left panels), June through August (middle panels) and September through November (right panel). Upper panels show the absolute difference between the 3-month period in 2018-2019 and 2020 in Dobson units. Bottom panels show the ratio between the 3-month period in 2018-2019 and 2020. Values denoted in bottom right of each panel are area-averaged difference within a 50 x 50km area around Manhattan (black box). 13 March – 29 April 2019 data are double counted in the March through May 2018 – 2019 period due to unavailable data in the 13 March – 29 April 2018 timeframe.**




These abrupt spatiotemporal changes in TCNO$_2$ detected by TROPOMI were remarkably consistent with the higher
resolution measurements from the ground-based Pandora network. Prior to lockdown, TCNO$_2$ in Manhattan and
Queens, NY, was characterized by high variability, often surpassing 2 DU (Fig. 3). NO$_2$ total columns in New
Brunswick, NJ, and New Haven, CT, were overall considerably lower than measurements in New York City, in
agreement with pre-pandemic TROPOMI retrievals (Table 1, Figs. 1, 3). Across all sites, pre-pandemic TCNO$_2$
showed a clear seasonal cycle typical of Northern Hemisphere mid-latitude locations, with maxima occurring during
the winter (Figs. 3, 4) due largely to increased fossil fuels for domestic heating, the longer tropospheric NO$_2$ lifetime

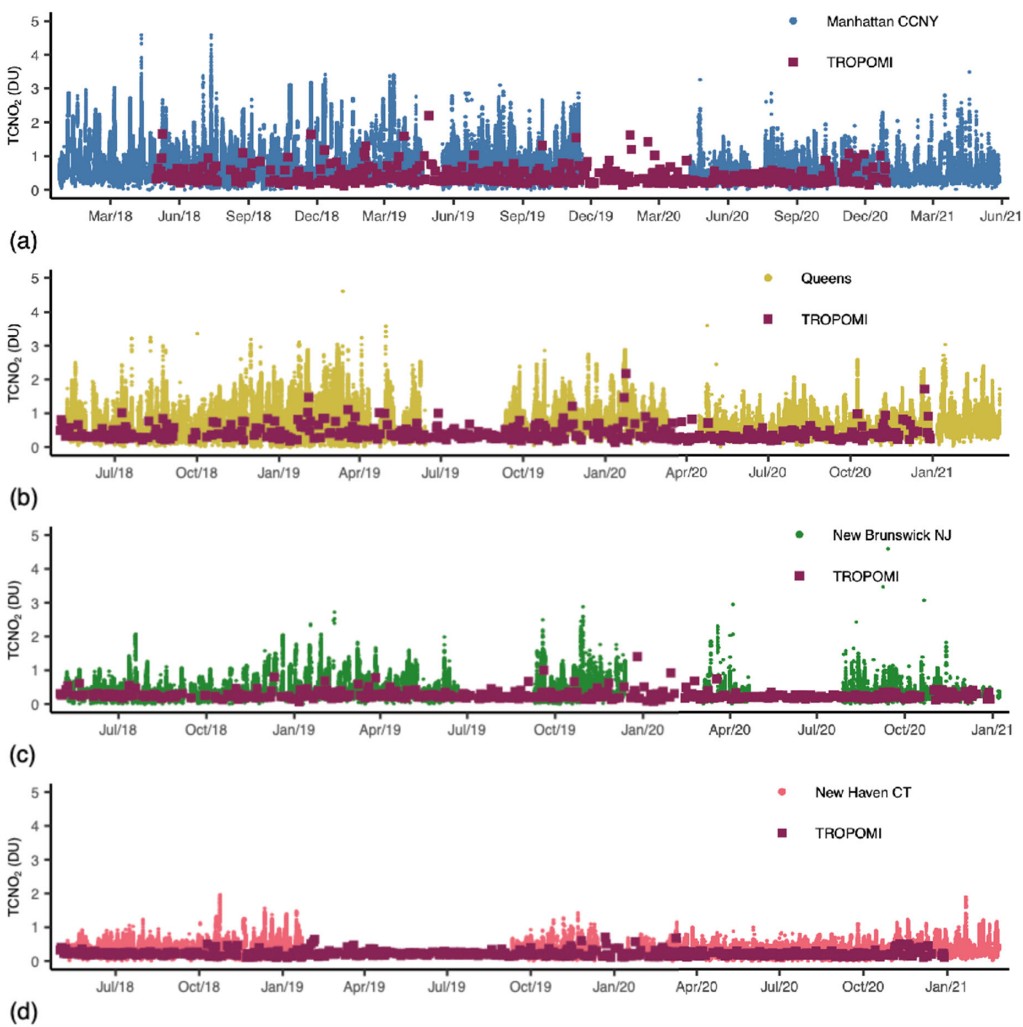

**Figure 3: Long term (December 2017- February 2021, May 2021 in Manhattan only) data record of TCNO$_2$ (in DU) measured by Pandoras in (a) Upper West Side Manhattan (blue circles), (b) Queens (yellow circles), (c) New Brunswick NJ (green circles) and (d) New Haven CT (pink circles). Total column TROPOMI overpass data at locations of the Pandora instruments is also shown (red squares). No data averaging was performed on Pandora or TROPOMI values.**



at colder temperatures, less light availability, and a shallower and more stable planetary boundary layer (A et al., 2008;
Semple et al., 2012). Post-shutdown, all Pandora sensors measured a significant drop in $TCNO_2$. In the two months
following the initial lockdown, $TCNO_2$ in Manhattan decreased by 36% compared to pre-pandemic levels, with
smaller declines, 21%, 19% and 29% respectively, in Queens, New Brunswick, and New Haven (Table 1). Variability
in $TCNO_2$ (Table 1) also decreased, indicating a reduction in the magnitude of high $NO_2$ pollution episodes. As social
distancing restrictions gradually started to ease in June, $TCNO_2$ in Manhattan started to slowly recover, reaching 25%
lower than the pre-pandemic seasonal mean in summer and 22% lower in fall 2020. $NO_2$ rose even closer to pre-
pandemic levels in Queens, New Brunswick, and New Haven, showing less than 15% decline in summer and fall 2020
(Table 1), consistent with our TROPOMI results. $TCNO_2$ in Manhattan, however, dropped again significantly below
pre-pandemic levels during the second wave of the pandemic in late 2020 (Table 1, Fig. 4). The decline in $TCNO_2$
reached 39% in January 2021, consistent with both a decline in mobility (i.e., re-closing of businesses and transition
from in-person to online learning in many schools in the area; Fig. S1) as well as favorable meteorological conditions
for low $NO_2$ (discussed in section 3.4). As restrictions eased again, $NO_2$ levels rebounded to 11% and 21% below pre-
pandemic levels in April and May 2021, respectively, more than a year after the COVID-19 outbreak in the U.S.
(Table 1, Fig. 4).

These changes resulted in a departure from typical seasonal $NO_2$ behavior, maximum in winter and minimum in
summer, with instead a maximum in monthly mean $TCNO_2$ in July 2020 and two minima tightly linked to the two
pandemic waves in May 2020 and January 2021 (Fig. 4). In agreement with Gkatzelis et al. (2021), the $NO_2$ decrease
closely followed changes in the stringency of lockdown measures and particularly decreases in traffic, further
confirming the importance of the transportation sector as a source of NOx pollution in Manhattan. Still, as discussed
in the next section, other emission sectors also contributed significantly to the observed spatiotemporal changes in
$NO_2$ pollution over the New York metropolitan area during the multiple waves of the pandemic.

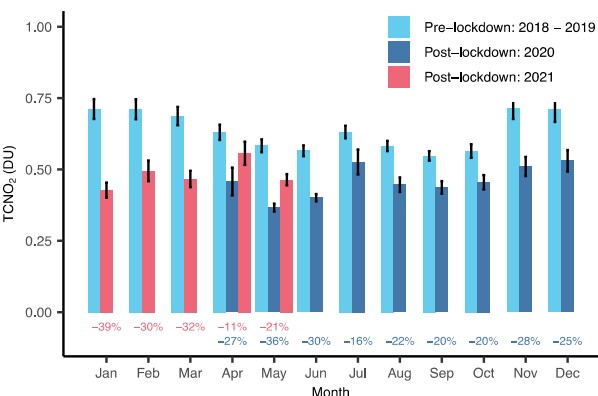

**Figure 4: Monthly mean seasonal cycle of TCNO₂ in Upper West Manhattan pre-lockdown (Dec 2017-Dec 2019, cyan) and post-lockdown (Apr 2020-Dec 2020, blue and January-May 2021, red), as measured by PSI #135 (30 min averaged data; 95% confidence intervals indicated by error bars; data not available during Jan-Mar 2020). The percent (%) change is also shown below each bar.**



## 3.2. Impacts of COVID-19 measures on NOx emissions

While meteorology plays a significant role in air pollution levels, our estimates of top-down $NO_x$ emissions from TROPOMI indicate that sudden reductions in NOx emissions due to COVID-19 measures were the dominant factor driving the observed $NO_2$ decline in New York City during the first wave of the pandemic (Fig. 5). Five-month (May to September) averaged top-down $NO_x$ emissions suggest a 34.5% drop between 2019 and 2020 (Fig. 5, right panel). This reduction in NOx emissions is significantly larger than the long-term decline of approx. 4% $yr^{-1}$ reported in previous studies for the eastern U.S. and New York City (Krotkov et al., 2016; Goldberg et al., 2019a), and suggests that COVID-19 measures during the first pandemic wave led to ~30% reduction in NOx emissions in New York City, on top of the long-term trend resulting from air-quality regulations and technological improvements. The reason $NO_2$ changes are smaller than NOx changes during the coincident timeframe ($\Delta NO_2$: ~24% vs. $\Delta NO_x$: ~35%) is because there is a background component to $NO_2$.

The EPA National Emissions Inventory (NEI) provides context for expected changes in NOx emissions due to the COVID-19 pandemic. According to 2017 NEI data, mobile sources account for about 59% of annual NOx emissions in New York City (25% on-road, and 34% non-road transportation including non-road equipment (15%) and locomotives/aircrafts/marine vessels (19%)). The next largest contributing sector is energy (41%), which includes electric generation, and residential, commercial, and industrial fuel combustion. Wildfires, biogenic sources, and waste disposal contribute a negligible amount (<1%; NEI 2017). New York City $NO_x$ emissions are more heavily weighted in the energy sector than other major U.S cities such as Los Angeles (13%) and Chicago (26%) (NEI, 2017). During spring 2020, MTA bridge and tunnel traffic decreased on average by 55%, nation-wide commercial passenger airline and business aviation travel decreased by approx. 75% and 70% (Transportation Research Board 2020; FlightAware 2020; Bureau of Transportation Statistics (BTS) 2020), while operation of commercial marine vessels,

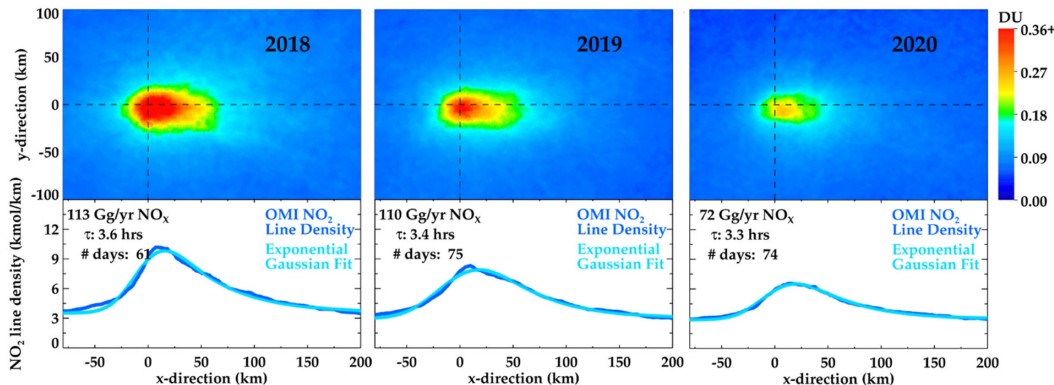

**Figure 5: Five-month averaged (May-September) top-down $NO_x$ emission estimates for the New York metropolitan area, for 2018 (left panel), 2019 (middle panel) and 2020 (right panel). TROPOMI $NO_2$ data is rotated based on daily wind direction. Bottom panels show the TROPOMI $NO_2$ line densities, which are integrals along the y-axis ± 50 km about the x-axis. The statistical EMG fit to the top-down line densities is shown in light blue.**



non-road equipment, and locomotives dropped by an estimated ~6%, ~45%, and ~15-20%, respectively (United
Nations Conference on Trade and Development2020; Procore, 2020; BTS 2020). These changes in mobility
correspond approx. to 26% change in NOx emissions. Declines in power generation demand/usage in New York City,
however, were considerably smaller, on average 15% in spring 2020 (New York Independent Systems Operator,
2020). These changes in emissions from the transportation and power generation sector suggest approximately 32%
decrease in NOx emissions in New York City during the first wave of the pandemic, which is consistent with our
estimated reduction in top-down NOx emissions from TROPOMI.

The overall less dramatic declines in $TCNO_2$ observed at locations outside Manhattan (e.g., CT and NJ) during the
first two months following the initial lockdowns agree with reported changes in population, with many city residents
across the US relocating (temporally and long term) to their suburban areas, more so from wealthier than lower-income
neighborhoods (Quealy et al., 2020). They are also consistent with mobility trends across our study region, with the
strongest mobility declines occurring in New York City. According to Apple mobility data, transportation associated
with driving and transit during March-May 2020 were 36% and 72% lower than baseline, respectively, in New York
City, compared to 32% and 54% in Middlesex County NJ and 19% and 49% in New Haven, CT (Fig. S3). Moreover,
the mobile sector constitutes a larger portion of total $NO_x$ emissions in Middlesex County NJ (72%) and New Haven
CT (71%) than in New York City, with significantly larger contributions from diesel at 36% of Middlesex total
emissions (22% in CT, 25% in NYC). National U.S diesel sales experienced a relatively smaller decrease from 2019
– 2020 than gasoline sales did, with a maximum decrease of ~10% in spring (vs. a mean -40% for gas) (U.S. Energy
Information Administration, 2021), so the relatively larger contribution from diesel in NJ could also partially explain
the smaller decreases in $NO_2$ at these locations compared to those observed in NY.
**3.3 Changes in NO₂ weekly cycles during the pandemic**
Anthropogenic $NO_x$ emissions often display a clear weekly cycle in major cities around the world, with minima on
rest days (e.g., Beirle et al., 2003; Kaynak et al., 2009; Tzortziou et al., 2013). The amplitude of this weekly cycle has
been shown in OMI data (2015-2017) to be strengthening in regions undergoing rapid emission growth, while it has
been weakening over European and U.S. cities due to the long-term decline in anthropogenic emissions (Stavrakou et
al., 2020). Yet, recent data from TROPOMI (2018-2019) show that large $NO_2$ column decreases on Sunday are still
prevalent in cities of North America, Europe, Australia, Korea and Japan (Stavrakou et al., 2020). In New York City,
TROPOMI captured 30% lower $NO_2$ on Sundays compared to a typical weekday in 2018-2019 (Goldberg et al., 2021),
in agreement with pre-pandemic MTA and Apple data showing lower traffic into and around the city on Sundays.
Similarly, Pandora measurements in Manhattan showed a clear weekly $NO_2$ dependence before the pandemic, with
minima consistently observed on Sunday on average 33% lower than weekday values (Figs. 6, 7). A strong diurnal
variability in $NO_2$ was also found (e.g., Fig 8), although diurnal patterns were highly variable spatially and temporally,
consistent with previous studies (Tzortziou et al., 2013). The Sunday-to-weekday column $NO_2$ ratio varied seasonally
from 0.64 and 0.63 in spring and summer, to 0.75 and 0.88, respectively, in fall and winter (Figs. 6, 7b), most likely





due to the longer tropospheric NO$_2$ lifetime and an increase in relative contribution of NO$_x$ sources that have no weekly
cycle (e.g., heating) in winter (Beirle et al., 2003).

The COVID-19 measures significantly impacted this weekly NO$_2$ behavior. Over the nine months following the
lockdown in New York (Apr-Nov 2020), TROPOMI captured a clear increase in the Sunday-to-weekday column NO$_2$
ratio from 0.76 to 0.92 (Fig. 7a). Higher frequency Pandora measurements enabled comparison on seasonal timescales,
revealing a disproportionate drop in weekday TCNO$_2$ immediately after the initial lockdown (Figs. 6, 7). Weekday
NO$_2$ decreased by as much as 36% and 29% in spring and summer 2020, respectively, while Sunday NO$_2$, decreased

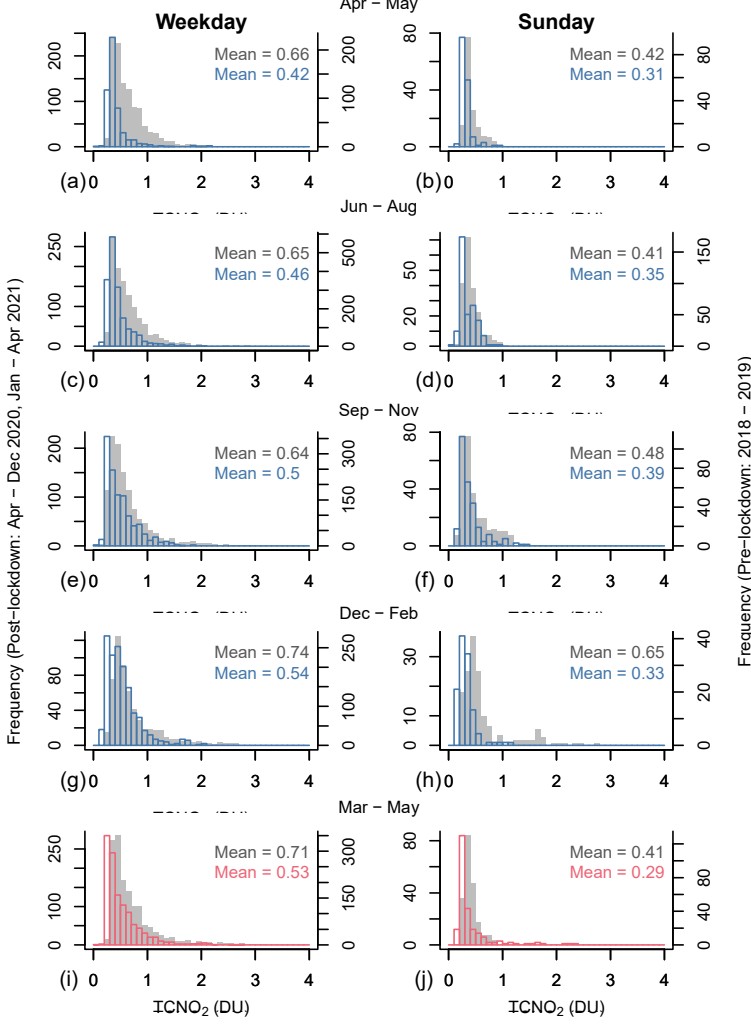

**Figure 6:  Histogram of TCNO₂ measured in Upper West Manhattan by PSI#135 for pre-lockdown (grey, 2018-2019) and post-lockdown winter (blue) and post-lockdown spring (pink) conditions. Results are shown for weekdays (left column) and Sunday (right column) across seasons from April 2020 to May 2021. The mean NO₂ pre- and post-lockdown is also shown.**



only by 26% and 15% (Fig. 6). The Sunday-to-weekday column $NO_2$ ratio, thus, increased by 16% in the post-
pandemic spring months with a similar trend into the summer (Fig. 7b). By fall, although $TCNO_2$ was still significantly
lower than pre-pandemic levels (-22% on weekdays and -19% on Sundays, Fig. 6), the typical weekly cycle re-
emerged with a post-pandemic ratio of 0.78. Surprisingly, the weekly cycle in $TCNO_2$ increased during the winter
(Fig. 7b), as a result of a larger decrease in Sunday $NO_2$ (49%) compared to weekday $NO_2$ (27%, Fig. 6). A large
departure from typical weekend travel patterns during the second wave of the pandemic, with MTA bridge and tunnel
traffic data showing a relatively larger decrease in traffic on Sundays during winter 2021 (Fig. S3), could partly explain
these results while the adoption of socially distanced protocols by 2021 may have resulted in relatively fewer
reductions of weekday activities such as construction or shipping. By the reopening phase in March-May 2021, the
weekly $NO_2$ cycle strengthened significantly (Fig 7b). With the exception of two Sundays in March and April that
showed high peaks in $TCNO_2$ due to strong influence of low-speed (<5 ms$^{-1}$) south and westerly winds, the Sunday-
to-weekday ratio approached pre-pandemic levels in spring 2021, likely reflecting a return to "normal" as the city-
wide COVID infection rate dropped (Fig. SI).

Long-term declines in anthropogenic $NO_x$ emissions and the resulting growing importance of background $NO_2$ had
already led to a significant dampening of the weekly $NO_2$ cycle in pre-pandemic New York City over the past 15
years, as shown by an increase in the OMI retrieved Sunday-to-week column ratio by 17% from 2005 to 2017 (Qu et
al., 2021; Stavrakou et al., 2020). Interestingly, the early stringent COVID-19 lockdown measures and related abrupt
changes in human behavior resulted in an additional 16% weakening of the $NO_2$ weekly cycle, in just three months.
Including these changes (both weakening and recovery) in weekly cycles of emissions and pollutant concentrations in
chemistry-transport models is important in efforts to quantify and simulate the impacts of the COVID-19 pandemic
on regional air quality, human health, and ecosystems.

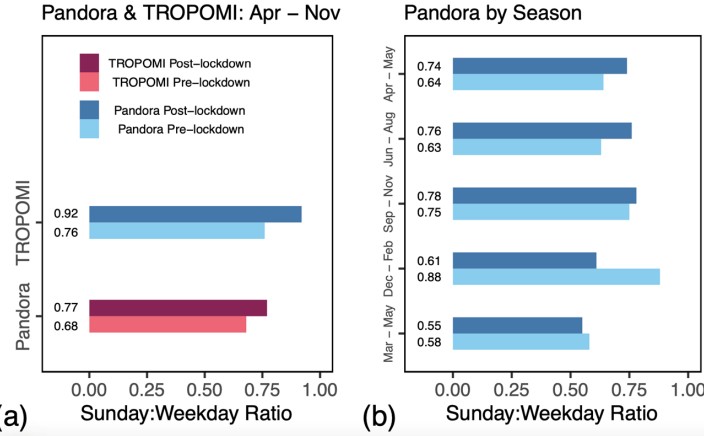

**Figure 7: (a) Sunday-to-weekday TCNO₂ ratios averaged over Apr-Nov 2018–2019 (pre-lockdown) and 2020 (post-lockdown) from TROPOMI and Pandora (PSI#135); (b) Seasonal change in Sunday-to-weekday column ratios pre- and post-lockdown from Pandora (PSI#135).**



### 3.4. Meteorology as a driver of NO₂ decline and high pollution episodes during the pandemic

Despite the significant reduction in $NO_2$ emissions during and following the COVID-19 lockdown, both ground-based and satellite sensors captured cases of high pollution in the New York metropolitan region with column $NO_2$ often exceeding three times the pre-pandemic levels (Table 1, Figs. 4, 8). April 23 and 25, 2020, during the initial lockdown, are such instances of $TCNO_2$ exceeding 1.8 DU (three times the pre-pandemic seasonal $NO_2$ mean) and showing remarkably similar diurnal behavior at the Manhattan and Queens locations (Figs. 8c, d). TROPOMI data was not available, but OMI captured $NO_2$ > 1 DU over New York city on April 25 (data not shown here). At the early stage of the second wave of the pandemic, $TCNO_2$ also exceeded 1.8 DU on October 9 in both Manhattan and Queens with a time-lag of approximately 2 hours between the maximum observed by the two instruments (Fig. 8e). On the same day, TROPOMI $TCNO_2$ reached 0.9 DU, more than two times higher than the pre-pandemic satellite monthly $NO_2$ mean (Fig. 8e). Overall, there were 12 days when ground-based measured $TCNO_2$ exceeded 1.8 DU in post COVID-19 New York City, despite a 34.5% drop in top-down NOx emissions (Fig. 5). Considering the significant decline in transportation emissions, the post-lockdown high $NO_2$ pollution episodes are most likely associated with power plant emissions and specific meteorological conditions. Indeed, the EDGAR v5.0 inventory shows that spatial patterns in NOx emissions over the New York metropolitan area are primarily driven by the power generation sector, while contributions from road traffic, buildings, and manufacturing show more even distribution with slight peaks of

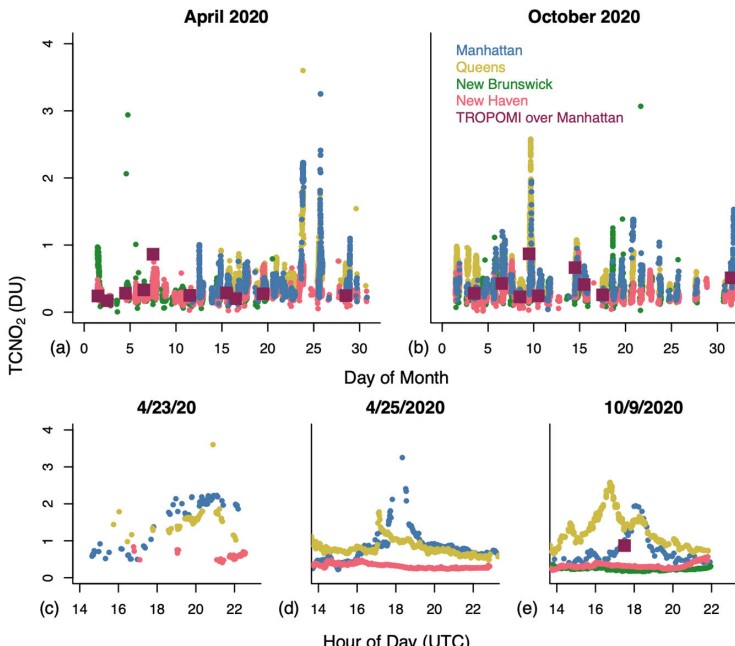

**Figure 8: Despite the decline in traffic and physical distancing restrictions, cases of exceedances (TCNO₂ > 1.8 DU) were observed in the New York metropolitan area during and post the COVID-19 lockdown. TCNO₂ measurements are shown here for (a) April 2020 and (b) October 2020, from TROPOMI and Pandora systems in Manhattan, Queens, New Brunswick, and New Haven. Diurnal dynamics in TCNO₂ during specific days of exceedances are shown for (c) April 23, (d) April 25 (no TROPOMI data available), and (e) October 9.**





approximately 0.05 kg $NO_x$ $m^{-2}yr^{-1}$ in Brooklyn, Queens, and Manhattan (Fig. 9). Among the many power plants in
the area, the Astoria Energy LLC and Astoria and Ravenswood Generating Stations in Queens were among the largest
greenhouse gas polluters in the state of NY in 2018 and 2019, with total reported greenhouse emissions >3,500,000
metric tons $CO_2e$ (EPA FLIGHT GHG Inventories) (Fig. 1). In NJ, the PSEG Bergen Generating Station in Ridgefield
(NW of Manhattan/NW of Harlem) and the Linden Cogeneration Facility (SW of Manhattan) are major power plants
located West of Manhattan with total reported emissions >7,000,000 metric tons $CO_2e$ both in 2018 and 2019.

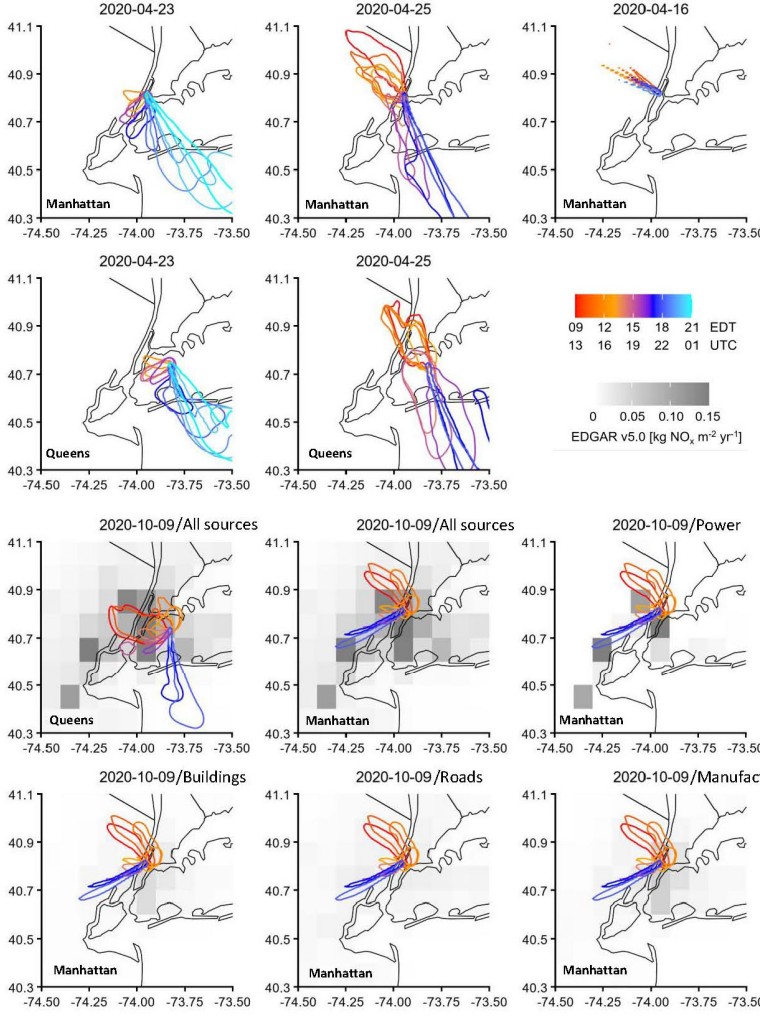

**Figure 9: Twenty-four hour total STILT surface influence contours for total column NO₂ exceedances on (a) April 23, (b) April 25, and (c) October 9, 2020, and a low NO₂ case on (c) April 16, 2020 for comparison. Contour lines represent surface influence of 1 ppb (μmol m⁻² s⁻¹)⁻¹and are colored by hour-of-day of the receptor. October 9 is overlaid with EDGAR inventories of NOₓ for 2015 (kg NOₓ m⁻²yr⁻¹). The area encircled by each contour indicates the region of emissions that reaches the Manhattan and Queens observation sites for a given time and day.**





Consistent with the location of these power plants, we found that meteorological conditions on days when high $NO_2$
was measured in Manhattan were characterized by low-speed southerly and westerly winds. STILT footprints showed
that on April 23 air masses from the high-emitting power sector in NJ and along the East River persisted over Upper
West Manhattan from 1600 to 2100 UTC (Fig. 9) when $TCNO_2$ peaked in PSI #135 observations (Fig. 8c). A strong
increase in wind speed and change in direction, effectively mixing in clean ocean air, after 2100 UTC coincided with
a rapid decline in measured $TCNO_2$. A similar pattern was observed on April 25 (Figs. 8d, 9), when air intercepted by
the Manhattan and Queens Pandoras shifts from the NW to SE, slowing while passing over NJ and the East River
power plants around 1800 UTC to produce the observed $TCNO_2$ peak at these sites. On October 9, westerly airflow
from NJ shifted to accumulate NOx emissions over the Manhattan Pandora location from 1700 to 1900 UTC when
observed $TCNO_2$ peaked at 1.95 DU. Wind accelerated and shifted SW in the evening, coinciding with a $TCNO_2$
decrease to <0.5 DU (Figs. 8e, 9). Low-speed westerly winds brought Manhattan and East River power plant emissions
to the Queens location approximately 2 hours earlier that day, in agreement with the earlier peak in $TCNO_2$ measured
by the Pandora (Fig. 8e). Strong winds, persisting in a single direction for several hours, consistently dispersed
pollution resulting in low $NO_2$ column amounts over Manhattan and Queens. An example is April 16 (Fig. 9), when
high-speed NW winds persisted throughout the day dispersing local and regional pollution and transporting $NO_2$ out
to the ocean.

Combining the STILT footprints, which account for the meteorology described above, with the sector-specific
EDGAR NOx emission maps allows us to approximate the fraction of expected NOx concentration enhancements
from each emission sector observed at each Pandora station. For 25 April, we find that the largest contribution of NOx
at the Manhattan site is from power generation (42%), with manufacturing dominating at the Queens site (30%). Road
transportation (using pre-pandemic estimates) contributes only 13% and 18% at the Manhattan and Queens sites,
respectively. Notably, despite the constant emissions rate in EDGAR, the diurnal pattern in near-surface total NOx
concentration enhancement matched well with the observed $TCNO_2$ observed by the Pandoras on both 23 and 25 April
(not shown here). This result supports the large role played by meteorology in causing $NO_2$ accumulation and

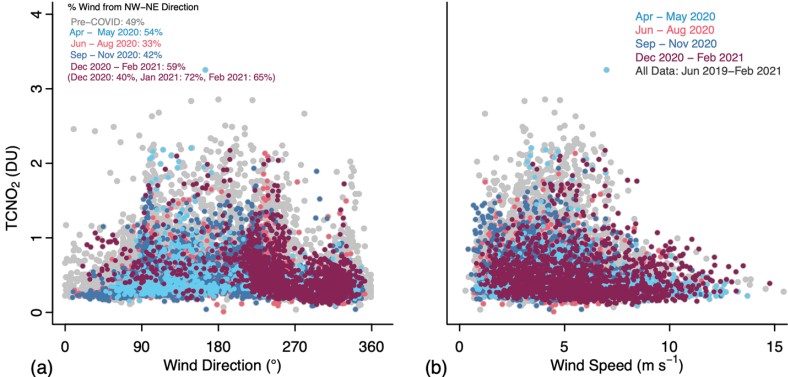

**Figure 10: Relationship between column NO₂ amounts (DU, PSI #135 data, 15-min averaged) and (a) wind direction (in degrees from north) and (b) wind speed (in m s⁻¹; ATMOS 41 data) organized by post-lockdown season, measured from June 2019 to February 2021 in Upper West Manhattan.**



demonstrates a clear connection between the near-surface and total column NOx concentrations on these days.

Our measurements showed that the observed correlation between particularly high post-pandemic $NO_2$ pollution
episodes and low-speed winds is typical of $NO_2$ dynamics in Manhattan. In large cities with relatively flat topography,
including New York City, increasing wind speeds from nearly stagnant to >8 m s$^{-1}$ were previously shown to decrease
$NO_2$ by 40–85% (Goldberg et al., 2020). Indeed, coincident measurements of wind conditions and $NO_2$ at the
Manhattan Pandora location before the pandemic showed that $TCNO_2$ rarely rose above 1 DU at wind speeds faster
than 8 m s$^{-1}$ (Fig. 10b). The highest $TCNO_2$ amounts occurred when surface winds were in the range 1-5 m s$^{-1}$. Under
such conditions, winds are strong enough to transport pollution from local sources as well as major pollutant emitters
in the tri-state area but can still lead to accumulation of pollution in Manhattan.
Moreover, the frequency of high $NO_2$ pollution events varies by wind direction, which correlate with sources of NOx
pollution. Most events with $TCNO_2$ > 1 DU, and all cases with $TCNO_2$ > 2DU, occurred with SE-SW winds (90-270°
in Fig. 10a). These air mass origins encompass influences from Queens and Brooklyn (SE), lower Manhattan, and
northern New Jersey (SW-W) where most of the major power plants and economic activity are located (Fig. 1). Mean
$TCNO_2$ for SE-SW winds was 0.6 DU, compared to 0.4 DU for NE-NW winds. Pre-pandemic TROPOMI retrievals
(2018-2019) also showed that SE-SW winds yield the highest $NO_2$ levels in New York City, on average twice as high
compared to winds from the NW and NE (Fig. 11), where there are fewer upwind sources. Satellite imagery over the
2018-2019 period was evenly distributed across SE-SW (high $NO_2$) and NW-NE (low $NO_2$) wind directions.
TROPOMI retrievals also demonstrate a strong negative relationship between satellite $NO_2$ columns and wind speed
(Fig. 12), with the highest $NO_2$ occurring at wind speed < 4 m s$^{-1}$ and the lowest at wind speed >6 m s$^{-1}$ over the New
York metropolitan area before the pandemic.

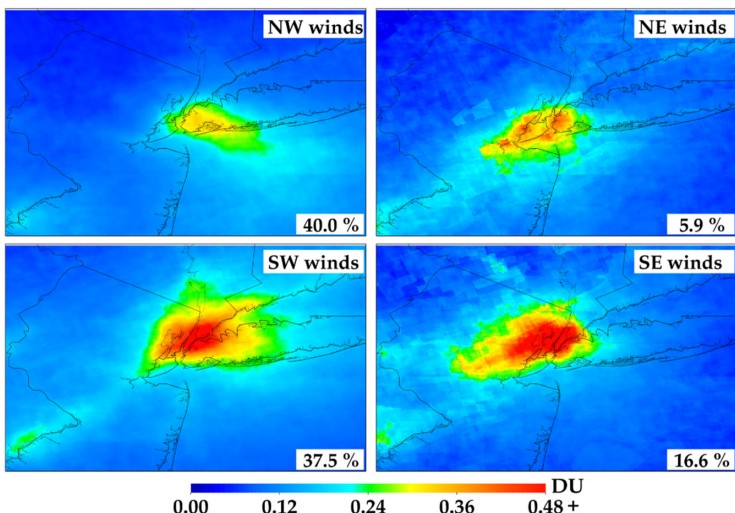

**Figure 11: Interpolated TROPOMI NO$_2$ plumes over New York City in 2018-2019 segregated into 100-m wind direction quadrants NW (top left), NE (top right), SW (bottom left), and SE (bottom right). The percentages of each direction are shown at the bottom right corner of each panel.**





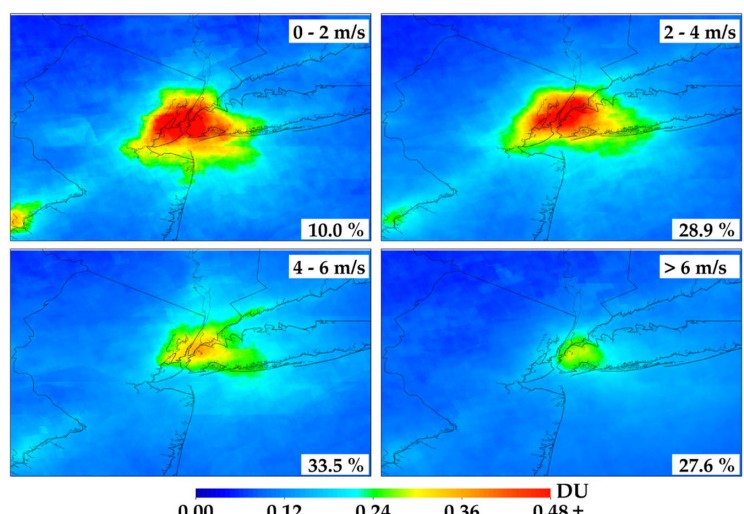

**Figure 12: TROPOMI NO$_2$ segregated by 100-m wind speed in 2 m s$^{-1}$ intervals from ERA5 daily meteorology. The percentages of each wind-speed interval are shown at the bottom right corner of each panel.**


These meteorological factors, in addition to explaining the particularly high TCNO$_2$ values measured even under strict
social distancing restrictions during the COVID-19 lockdowns in the tri-state area, were also found to contribute to
the significantly reduced NO$_2$ values in winter 2021. January and February 2021 showed a drop in NO$_2$ by 39% and
30%, respectively, similar to the NO$_2$ decline observed immediately after the initial strict lockdowns (Fig. 4). Although
traffic (based on both MTA data and Apple mobility trends) showed a noticeable decrease during the second wave of
the pandemic, mobility was not nearly as restricted as in April-May 2020 (Figs. S1-S3). Bridge and tunnel traffic was
approximately 30% lower in winter 2021 compared to 55% lower in spring 2020. Interestingly, in winter 2021 wind
in Upper West Manhattan was mostly (72% in January and 65% in February) from NW-NE directions, which yields
the cleanest conditions and favors low NO$_2$ columns (Fig. 10a). For comparison, wind at the same location in January
and February 2020 was 49% and 50% from NW-NE direction. In contrast to winter 2021, in spring, summer, and fall
2020, wind was 54%, 33% and 42% from NW-NE directions (compared to 49% in pre-covid conditions, Fig. 10) and
mean wind speed was in the range 3.8-5.5 m s$^{-1}$, suggesting that wind conditions were not favorable for lower NO$_2$ in
Manhattan in 2020. Hence, our estimates of NO$_2$ decline in April-December 2020 primarily reflect the impact of
changes in anthropogenic emissions, particularly reductions in emissions from the transportation sector. These
findings corroborate results from Goldberg et al., (2020), who concluded that varying meteorological conditions (wind
speed and direction) in New York City, while different between years, did not have a strong biasing effect in their
estimates of the effects of COVID-19 physical distancing on NO$_2$ in the month directly following the initial
lockdowns. The prevalence of northerly winds in winter 2021, however, minimized the relative contribution of
emissions from the energy sector to New York City, favoring low NO$_2$ conditions. This led to stronger NO$_2$ declines
compared to pre-pandemic levels than would be expected based on just changes in emissions from the transportation
sector during the second wave of the pandemic.


### 4. Summary and conclusions

Stringent lockdown measures following the COVID-19 outbreak resulted in an abrupt and significant decline in TROPOMI top-down NOx emissions in New York City, by ~30% on top of long-term trends. A sudden drop in total column $NO_2$ (by up to 36% in Manhattan), along with a weakening of the weekly $NO_2$ cycle and a disruption of typical seasonal patterns were observed by the ground-based Pandora network in the New York metropolitan area. Yet, during the same timeframe, traffic in New York City bridges and tunnels plummeted by 55%, on average, compared to pre-pandemic levels, reaching as much as 80% reduction in early April 2020. These results highlight that although on-road transportation is an important source of emissions in New York City, emissions from non-road transportation and the power generation sector (not as strongly affected by the lockdown measures) critically affect $NO_2$ pollution levels in New York. Accounting for each sector's contribution to total emissions, resulted in a change in NOx emissions by approx. 32%, which was consistent with satellite top-down estimates.

Disentangling the impacts of meteorology and NOx emission changes on urban air quality is key for designing and implementing improved emission-control strategies. Meteorology had different impacts across the different pandemic waves in New York City. Although it was not found to have a strong biasing effect after the first pandemic wave in spring to fall 2020, meteorology strongly favored clean air conditions over Manhattan after the second pandemic wave in winter 2021, lowering $NO_2$ levels beyond what would be expected based on lockdown measures alone. The key role that meteorology plays in shaping the relative contributions from different emission sectors to $NO_2$ pollution in New York City was further demonstrated by the occurrence of several high $NO_2$ pollution events even during – and despite - the extreme reductions in transportation emissions during the stringent early lockdowns. High $NO_2$ columns, often exceeding three times the pre-pandemic levels, were consistently characterized by low-speed (< 5m s$^{-1}$) SW-SE winds that enhanced contributions from the high-emitting power-generation sector and accumulation of pollution over New York City. A subsequent increase in wind speed and change in wind direction typically coincided with a decrease in $NO_2$ over the city, indicating dispersion of pollutants across the coastal environment with potentially negative effects on downwind communities as well as terrestrial and aquatic ecosystems (Loughner et al., 2016).

The COVID-19 pandemic resulted in immediate and multifaceted impacts on human behavior that affected various pollutant sectors and their relative contributions to urban NOx emissions differently. During this extreme natural experiment, long-term and high-temporal resolution retrievals from the Pandora network were essential in capturing the response of column $NO_2$ – declines and high pollution episodes - during the multiple pandemic waves and reopening phases in the New York metropolitan area. Incorporating observed NOx emissions changes across timescales is important for improving air quality modeling and forecasting, especially in the context of sub-daily stagnation events that produce NOx exceedances despite low emissions. Such high-resolution observations from ground-based networks, and soon from geostationary satellite sensors such as TEMPO (Chance et al., 2013), enable the characterization of fine-scale features in $NO_2$ behavior as well as assessment of the possible effects of rapid meteorological changes on air quality conditions. In New York, a city transitioning to a NOx limited ozone production environment during summer (Jin et al., 2017), NOx plays an important role in the oxidation of VOC's ozone


production as well as secondary aerosol formation. Integration of high-resolution $NO_2$ measurements from ground-
based networks and geostationary satellite platforms is, thus, critical in further assessing changes in $NO_2$, aerosol, and
ozone pollution as the world re-opens, and in evaluating the effectiveness of future sector-specific NOx emission
control strategies and their impacts on air quality, human health, and urban ecosystems.
**Acknowledgements:** We thank Alexander Cede, Thomas Hanisco, Moritz Mueller, Michael Gray, Elena Spinei Lind,
Brian Lamb and the NASA Pandora Project and ESA Pandonia Project staff for assistance in the field and for
establishing and maintaining the Pandora sites used in this investigation. We also thank Rohit Mathur and Venkatesh
Rao for feedback on an earlier draft of this manuscript. This research was supported by NASA Rapid Response and
Novel research in the Earth Sciences (RRNES) Program, grant number 80NSSC20K1287, NASA Interdisciplinary
Science (IDS) Program, grant 80NSSC17K0258, NOAA Earth System Sciences and Remote Sensing Technologies
award NA16SEC4810008, and NOAA Climate Program Office's Atmospheric Chemistry, Carbon Cycle, and Climate
program, grant number NA20OAR4310306.
**Disclaimer:** The research described in this article has been reviewed by the U.S. Environmental Protection Agency
(EPA) and approved for publication. Approval does not signify that the contents necessarily reflect the views and the
policies of the agency nor does mention of trade names or commercial products constitute endorsement or
recommendation for use.



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
