# Peer review of "Declines and peaks in NO2 pollution during the multiple waves"

_Atmospheric Chemistry and Physics, 2021_

## Author Response (AR1)

**Response to Reviewer Comments**

The authors focus on the declines and peaks in NO2 pollution during the multiple lockdown phases in the New York metropolitan area and disentangle the contribution of anthropogenic emissions sources and role of meteorology. In general, I find this manuscript to be of interest for publication and appropriate for Atmospheric Chemistry and Physics. I have one main concern and several minor suggestions for improvement listed below that should be considered by the authors before publication.

The authors use total column NO2 in this study and a few references are made to tropospheric column NO2 throughout the manuscript. The authors can be specific about whether the "column NO2" refers to "total column NO2" or "tropospheric column NO2". I have pointed out a few instances below. Typically, studies use tropospheric column NO2 to relate to changes in NOx emissions. How do the values of total column NO2 compare to tropospheric column NO2 for New York metropolitan area? The authors should consider discussing why total column NO2 is used and the possible impacts on the results.

*We thank the Reviewer for their comments; they have substantially improved our manuscript. We also thank the reviewer for finding this manuscript to be of interest for publication and appropriate for Atmospheric Chemistry and Physics. Below, we include our response to the Reviewer's comments.*

*Following the Reviewer's suggestion, we made changes throughout the document to be more specific about whether the "column $NO_2$" refers to "total column $NO_2$" or "tropospheric column $NO_2$". The Pandora direct-sun retrievals provide the total column amount of $NO_2$, and any retrievals of tropospheric column would have to be based on assumptions regarding the stratospheric contribution. TROPOMI $NO_2$ is reported as total (tropospheric + stratospheric) column $NO_2$ when we compare to Pandora results, for consistency and more direct comparisons. Since in polluted areas such as New York most of the $NO_2$ is from anthropogenic emissions, the stratospheric amount is relatively low, and not as highly variable (spatially or temporally), compared to the tropospheric $NO_2$ column amount. Any change in NOx emissions would manifest as a larger % change in tropospheric $NO_2$ amount than in total column $NO_2$ amount. As we mention in the revised manuscript, our estimates of total column $NO_2$ changes from TROPOMI (and Pandora) are, thus, smaller than our TROPOMI estimates of NOx changes during the same timeframe, because there is a background component to $NO_2$ (see lines 332-333).*

Minor comments

Line 68 Population and area of New York metropolitan area?

*Information is provided in the revised manuscript (lines 70 and 75).*

Line 81-86 Do both these studies use total column NO2? Or tropospheric column NO2?

*The sentence has been modified to clarify this. These studies only use tropospheric column $NO_2$.*

Line 85 "0.4° radius". Also mention in kms to compare to 100km radius in previous sentence.

*Revised. This area was a $0.4^o$ x $0.4^o$ box centered on New York City, which is approximately equivalent to a $0.2^o$ radius or approximately a 22 km radius of New York City.*

Line 101 "4% yr$^{-1}$ decrease …" is in tropospheric column NO2. How would the trend values be for total column NO2?

*Due to the relatively small amount of $NO_2$ in the stratosphere compared to the troposphere in urban areas like NYC, the difference is not expected to be large. For example, assuming ~ 0.1 DU stratospheric column $NO_2$ (Geddes et al, 2018; https://doi.org/10.5194/amt-11-6271-2018) for an average of 0.62 DU $TCNO_2$ (i.e., annual average $TCNO_2$ in Manhattan and Queens), this would correspond to ~approx. 3.5%/yr decrease.*

Line 106 "…highest national NO2 levels." The authors can give value of annual mean NO2 and compare to the recently updated WHO guidelines.

*Herman et al (2018) refer here to the total column $NO_2$ from PSI#135, while WHO reports surface levels $NO_2$. Results from PSI#135, including seasonal means pre- and post-pandemic, are discussed in more detail in this manuscript (section 3).*

Line 117 Here and everywhere else, the authors mention high-frequency observations from Pandora but do not provide any value.

*This information is now provided in the revised manuscript (section 2.1). The temporal resolution for Pandora measurements was approximately 1 minute.*

Line 131 Section heading can be changed to "Materials and Methods" as the subheadings also focus on the various datasets.

*To be more inclusive of different aspects of the methodology used here, we used Methods as the subheadings (i.e., datasets, study sites, approach, algorithms).*

Line 135 Last assessed date for the URL.

*Done. We included this information (PGN, 135 https://www.pandonia-global-network.org/, accessed June 4 2021) on Line 135 of revised manuscript.*

Line 169 Tropospheric columns of?

*This was referring to $O_3$, $NO_2$, $SO_2$ and $CH_2O$. We revised to "retrievals of $O_3$, $NO_2$, $SO_2$ and $CH_2O$ total columns and information on vertical profile".*

Line 173 Filtering criteria such as TCNO2>0 can lead to a positive bias in the mean TCNO2. Is it possible to quantify how much data is removed because of this filtering criteria and if the positive bias is large?

*$TCNO_2 < 0$ does not have a physical meaning, and in almost all cases corresponded to cases when the error in the measurements was high. Less than 1% of the data was removed due to this filter.*

Line 187 The authors use total column NO2, but the retrieval steps are also given for tropospheric vertical column.

*This is a good point by the reviewer. TROPOMI data is also reported as tropospheric column in this manuscript for direct comparison with previous studies. This was clarified in the revised manuscript.*

Line 189-190 Would the version change have a significant impact on the results?

*The change between v1.02 and v.1.03 is considered to be minor by algorithm developers. Erroneous data is better filtered using the qa_flag in version 1.03 as compared to version 1.02 resulting in "smoother" less pixelated long-term averages. Based on the Users' manual, we do not expect a significant bias due to this algorithm change.*

Line 193 Here again, the authors mention about validation of "tropospheric columns".

*This was clarified in the document.*

Line 253 "(v) in March-April 2021" should be "March-May 2021", right?

*Thank you for catching this, correction was made.*

Line 266 URL or DOI for MTA data?

*This information is provided now on Line 269 of revised manuscript (For MTA data (https://new.mta.info/coronavirus/ridership, accessed 4 June 2021).*

Line 297 "Variability in TCNO2 also decreased…" except New Brunswick?

*Thank you for catching this, correction was made.*

Line 320 "Fig.5, right panel." "Middle and right panels" perhaps?

*Good point, we revised to "Fig. 5" for brevity.*

Line 324-326 This statement starts suddenly and the value for NO2 changes need to be given before explaining why they are lower compared to NOx changes.

*To clarify, we revised this sentence: "The reason TROPOMI $TCNO_2$ changes (Fig. 2) are smaller than NOx changes during the coincident timeframe ($\Delta TCNO_2$: ~24% vs. $\Delta NOx$: ~35%) is because there is a background component to $NO_2$". The value for $NO_2$ change is given in the parentheses.*

Line 339-344 The authors can consider giving a brief description of how these changes in NOx emissions were obtained (either here or in section 2).

*This paragraph was revised to address the reviewer's comment. In the revised manuscript we mention that "Applying these reported changes in activity to corresponding estimated $NO_x$ contributions from different components of the mobility sector in New York City (EPA) results in an approx. 26% decrease in $NO_x$ emissions."*

Lines 484-492 Interesting result!

*Thank you.*

Table 1 The table caption can include details of the time. "Present" may be replaced by "05/2021" Additional footnotes can be used on the column of "Temporal range of data" to mention data unavailability. For example, data from PSI#135 not available for Jan-Mar 2020 is mentioned in Figure 4 and should also be mentioned in table 1. The value of stdev for PSI#56,#69 for Sept-Nov reads as "0.24x". Check for typo. Lastly, are both the PSIs at Queens, New Brunswick and New Haven in operation for the time period stated and is the data from both of them used?

*We made the changes following the reviewer's suggestions. The two PSIs at the Queens, New Brunswick and New Haven locations operated during different time periods and data from both are included in our study. Time periods when data at the four locations was not available is shown in Figure 3 that provides the full time series of TCNO$_2$ measurements for all sensors.*

Figure 1 Is there a way the different major pollutant emitters can be identified on the map? For example, by use of numbers with the circles and the numbers can be stated in the figure caption. This would aid the interpretation of the results on lines 419-424.

*We revised the figure to be able to easily identify the major pollutant emitters on the map.*

[Figure]

**Figure 1: Map of study area, indicating location of Pandora sensors (white symbols) in Manhattan NY, Queens NY, New Brunswick NJ, and New Haven CT, overlaid with mean 2019 annual total column NO$_2$ from TROPOMI (in DU). Major pollutant emitters (red circles) in the area are included, specifically the PSEG Bergen Generating Station in Ridgefield (BG), the Linden Generating Station (LG) and the Phillips 66 Bayway (PB) Refinery in Linden (major emission sources in NJ), and the Astoria (AG) and Ravenswood Generating (RG) Stations in Queens, NY (among the largest greenhouse gas polluters in the state of NY in 2018 and 2019).**

Figure 2 What is the "ratio difference"? It is not clear in the figure caption also.

*This was changed to "ratio".*

Figure 7 The colors for TROPOMI and Pandora bars in (a) are different from the ones in the key. In panel (b), please add the year next to the months and consider adding error bars.

*Done.*

[Figure]

**Figure 7: (a) Sunday-to-weekday TCNO₂ ratios averaged over Apr-Nov 2018–2019 (pre-lockdown) and 2020 (post-lockdown) from TROPOMI and Pandora (PSI#135); (b) Seasonal change in Sunday-to-weekday column ratios pre- and post-lockdown from Pandora (PSI#135).**

Figure 8 The caption can be slightly modified to reflect that TROPOMI observations are only over Manhattan.

*Done*

Figure 9 The authors can consider labeling the panels and refer to the individual panel in the text.

*Done*

Figure 12 The time and location are missing in the figure caption.

*These are showing averages between May 2018 and December 2019. This is applicable to both Figures 11 and 12. The figure captions were revised (Figs 11 and 12).*

**Response to Reviewer Comments**

**Declines and peaks in NO2 pollution during the multiple waves of the COVID-19 pandemic in the New York metropolitan area**

Maria Tzortziou, Charlotte F. Kwong, Daniel Goldberg, Luke Schiferl, Róisín Commane, Nader Abuhassan, James J. Szykman, Lukas C. Valin

This paper presents an extensive study of declines and peaks in $NO_2$ levels durng different phases in the COVID-19 pandemic in New York city. Columns of $NO_2$ from TROPOMI observations and PGN for four stations in the New York metropolitan area are presented for periods pre- and post-pandemic. These observations shown changes in $NO_2$ columns up 36%. Additionally, model simulations of air masses and meteorological information are combined with PGN and TROPOMI observations in order to evaluate impact of $NO_2$ pollution in the New York metropolitan area during the multiple waves of the COVI-19 pandemic.

The topic of this work fits well within the scope of ACP. The manuscript is well structured and well written. However, the main concern is about the lack of evidence of decrease solely to lockdown periods during pandemic but not to as a general reduction NO2 emissions in New York. Would be possible to evaluate this decrease during lockdown as consequence of only mobility reduction in New York city and not as result of decreasing tendency of general reduction of NOx emission as suggest other studies (e.g. Zangari et al., 2020) by comparing with earliest years?

I recommend acceptance to ACP after addressing the comments above and few minor comments below.

*We thank the Reviewer for finding the manuscript well-structured and well-written and for recommending acceptance to ACP after addressing their comments. Below, we include our response to the Reviewer's comments and how we addressed them in the revised manuscript.*

*To address the Reviewer's main comment regarding comparison with previous years, we included a supplementary figure in the revised manuscript. This figure (S3) shows the long-term change in tropospheric column $NO_2$ in New York City over the 2005-2019 period from satellite OMI observations. In the revised manuscript (lines 289-291), we discuss that our results "can be compared to long-term $NO_2$ trends from OMI (Fig. S3), which shows a ~3.8% yr-1 drop between 2005 and 2019. The abrupt $TCNO_2$ changes during the initial phase of the COVID lockdowns, occurring within a matter of days, were approximately equivalent to the drop seen over the prior 10-year period between 2009 and 2019." In addition, in section 3.2 we mention that the estimated 35% reduction in 5-month (May to September) averaged top-down $NO_x$ emissions between 2019 and 2020 from TROPOMI "is significantly larger than the long-term decline of approx. 4% $yr^{-1}$ captured by OMI (Fig. S3) and reported in previous studies in previous studies for the eastern U.S. and New York City (Krotkov et al., 2016; Goldberg et al., 2019a), and suggests that COVID-19 measures during the first pandemic wave led to ~30% reduction in NOx emissions in New York City, on top of the long-term trend resulting from air-quality regulations and technological improvements." (Lines 328-332).*

**Additional comments:**

L179-180: Please add reference for OMI and SCIAMACHY instruments.

*Done. We added reference for OMI (Levelt, P. F., Van Den Oord, G. H. J., Dobber, M. R., Mälkki, A., Visser, H., De Vries, J., Stammes, P., Lundell, J. O. V., & Saari, H. (2006). The ozone monitoring instrument. IEEE Transactions on Geoscience and Remote Sensing, 44(5), 1093-1100.*
*https://doi.org/10.1109/TGRS.2006.872333); and SCIAMACHY (Bovensmann, H., Burrows, J. P., Buchwitz, M., Frerick, J., Noël, S., Rozanov, V. V., Chance, K. V., & Goede, A. P. H. (1999). SCIAMACHY: Mission Objectives and Measurement Modes, Journal of the Atmospheric Sciences, 56(2), 127-150).*

L247: Similar to the comment above, have you checked the $NO_2$ amounts for earliest years than 2018 for PGN or any satellite observation?

*To address the Reviewer's comment, we included a Supplementary figure in the manuscript (revised figure S3), showing the decrease in tropospheric column $NO_2$ in New York City over the 2005-2019 period from satellite OMI observations. The estimated change is a decline in $NO_2$ by 3.8%/year. These results are consistent with previous studies showing a long-term decline of approx. 4% $yr^{-1}$ for the eastern U.S. and New York City (Krotkov et al., 2016; Goldberg et al., 2019a). This new information and new supplementary figure are included in our revised manuscript, and changes during and after the COVID-19 pandemic are compared to long-term changes.*

**Figure S3: Annual average OMI tropospheric NO2 column, oversampled to 0.1 x 0.1 degrees, after filtering out data affected by the row anomaly, cloud fractions > 0.3, surface albedo > 0.3, solar zenith angles > 80 degrees, and the 5 rows at the edges of the swath that contain very large pixel sizes. The New York City spatial average shown here is equivalent to the box shown in Figure 2.**

[Figure]

L273: Would possible to generate a similar figure to SI-2 but comparing with 2018?

*Unfortunately, 2018 data is not available for New York City, therefore we only included comparisons with 2019.*

L276: Are you comparing tropospheric NO2 columns with total columns from ground based measurements? Or are both tropospheric columns?

*The TROPOMI results in Figure 2 are total (tropospheric and stratospheric) column amount for direct comparison with the ground-based Pandora observations. We decided to report total column amounts here because the Pandora direct-sun retrievals provide only the total (tropospheric and stratospheric) column amount of $NO_2$, and any retrievals of tropospheric column would have to be based on assumptions regarding the stratospheric contribution.*

L284: Would be possible to repeat the same figure for OMI or GOME-2 NO2 observations but including earliest years to confirm real reduction to COVID-19 lockdown and not a combination of other factors?

*To address the Reviewer's comment, we included a Supplementary figure in the manuscript (Figure S3, also shown above) showing the decrease in tropospheric column $NO_2$ in New York City over the 2005-2019 period from satellite OMI observations. As discussed in the paper, the NOx emissions reduction between 2005 – 2019 is approximately ~4% $yr^{-1}$ with perhaps a slight slowing in recent years (Goldberg et al., 2021). The changes we are showing here on sub-annual timeframes, and especially in the first 2-3 months following the lockdown, far exceed 4% $yr^{-1}$.*

L288: Already Manhathan and Queens present a significan reuction for 2018 in comparison 2019 before pacdemic lockdown, e.g. June-September and October 2018 and January 2019?

*The differences between fall 2018 and winter 2019 in the Manhattan and Queens $TCNO_2$ values are because of the seasonal patterns in $TCNO_2$ typically showing maxima in winter due largely to a combination of increased fossil fuels for domestic heating, the longer tropospheric $NO_2$ lifetime at colder temperatures, less light availability, and a shallower and more stable planetary boundary layer. Our estimates of five-month (May to September) averaged top-down NOx emissions from TROPOMI showed less than 3% decrease from 2018 to 2019, compared to a 34.5% drop between 2019 and 2020 (Figure 5 in revised manuscript).*

L290: Figure 3, TROPOMI columns are very low and difficult to observe the reduction in NO2 columns. What is the detection limit of NO2 trospospheric columns from TROPOMI?

*To address the Reviewer's comment, we changed both New Haven and New Brunswick Y-axis to Max 3 DU (not 5 as in Queens + Manhattan). We also changed the symbol used for the TROPOMI data to make the figure more readable. The uncertainty of TROPOMI slant column measurements is approximately 0.02 DU, and here we are showing values in the range of 0.2 – 2 DU, which would represent up to an uncertainty of 1-10% in the slant column.*

L331: Can be possible to quantify the increase or decrease of Nox emission due to only energy sector? It is clear that contribution from transportation sector is reduced, but energy sector, specially dosmestic consumption probably has an significant increase due to the COVID-19 lockdown.

*This is a very good point by the Reviewer. As we mention in the revised manuscript (lines 346-353), based on published reports (New York Independent Systems Operator, 2020), power generation demand/usage declined in New York City during the lockdowns but, on average, by only 15% in spring 2020 (compared to the much larger declines in traffic, for example MTA bridge and tunnel traffic decreased on average by 55%). Based on these numbers, and the 2017 NEI data showing that the energy sector contributes 41% of annual NOx emissions in New York City, we estimated that changes in emissions from the transportation and power generation sector would correspond approximately to 32% decrease in NOx emissions in New*

*York City during the first wave of the pandemic, which is consistent with our estimated reduction in top-down NOx emissions from TROPOMI.*

L340: Decrease of power demand/usage in New York city is associated to?

*This was mostly due to closing of high-energy consuming businesses and a large number of people moving out of New York City especially during the months immediately after the COVID-19 lockdowns were imposed.*

Figure 6: Please check the title display of subfigures.

*Done. A revised Figure 6 is included in the revised manuscript.*

Line 409: It would be worth to add a figure showing this data in the supplementary material.

*To address this comment, we revised Figure 8. The exact value from OMI is provided now in the text and in Figure 8.*

L427: Would be possible to add the location of the power plants in the maps similar to Figure 1?

*Done. The locations of power plants are now provided in revised Figure 1.*

[Figure]

**Figure 1: Map of study area, indicating location of Pandora sensors (white symbols) in Manhattan NY, Queens NY, New Brunswick NJ, and New Haven CT, overlaid with mean 2019 annual total column $NO_2$ from TROPOMI (in DU). Major pollutant emitters (red circles) in the area are included, specifically the PSEG Bergen Generating Station in Ridgefield (BG), the Linden Generating Station (LG) and the Phillips 66 Bayway (PB) Refinery in Linden (major emission sources in NJ), and the Astoria (AG) and Ravenswood Generating (RG) Stations in Queens, NY (among the largest greenhouse gas polluters in the state of NY in 2018 and 2019).**

L448-448: Please add a figure if it is possible in the supplementary material for the matched of TCNO2 in 23 and 25 April.

*To address this comment, we revised Figure 8, to show the agreement between the diurnal variability measured in TCNO2 and the EDGAR power-sector NOX enhancements in Manhattan. The text was also revised to: "Despite the constant NOx emissions rate for each month in EDGAR (i.e., no diurnal cycle), the diurnal pattern of the meteorology-driven simulated power-sector near-surface NOx concentration enhancement was consistent with the TCNO2 observed by the Pandoras on both April 23 and 25, 2020 (Fig. 8(c)-(e))"*

[Figure]

**Figure 8: Despite the decline in traffic and physical distancing restrictions, cases of high NO$_2$ pollution (TCNO$_2$ > 1.8 DU) were observed in the New York metropolitan area during and post the COVID-19 lockdown. TCNO$_2$ measurements are shown here for (a) April 2020 and (b) October 2020, from Pandora systems in Manhattan, Queens, New Brunswick, and New Haven, and TROPOMI over Manhattan. Diurnal dynamics in TCNO$_2$ during specific days of exceedances are shown for (c) April 23, (d) April 25 (square indicates OMI TCNO$_2$ over Manhattan), and (e) October 9, 2020. The EDGAR power-sector near-surface NOx concentration enhancements in Manhattan are shown by the grey line in (c)-(e).**

**References:**

Shelby Zangari, Dustin T. Hill, Amanda T. Charette, Jaime E. Mirowsky, Air quality changes in New York City during the COVID-19 pandemic,
Science of The Total Environment, Volume 742, 2020, https://doi.org/10.1016/j.scitotenv.2020.140496.
**Citation**: https://doi.org/10.5194/acp-2021-592-RC2